# Adverse Events in Anti-PD-1-Treated Adjuvant and First-Line Advanced Melanoma Patients

**DOI:** 10.3390/cancers16152656

**Published:** 2024-07-26

**Authors:** Daan Jan Willem Rauwerdink, Olivier van Not, Melissa de Meza, Remco van Doorn, Jos van der Hage, A. J. M. van den Eertwegh, John B. Haanen, Maureen J. B. Aarts, Franchette W. P. J. van den Berkmortel, Christiaan U. Blank, Marye J. Boers-Sonderen, Jan Willem B. de Groot, Geke A. P. Hospers, Djura Piersma, Rozemarijn S. van Rijn, A. M. Stevense-den Boer, Astrid A. M. van der Veldt, Gerard Vreugdenhil, Michel W. J. M. Wouters, Karijn P. M. Suijkerbuijk, Ellen Kapiteijn

**Affiliations:** 1Department of Dermatology, Leiden University Medical Center, Leiden University, Albinusdreef 2, P.O. Box 9600, 2300 RC Leiden, The Netherlands; d.j.w.rauwerdink@lumc.nl (D.J.W.R.); o.j.van_not@lumc.nl (O.v.N.); r.van_doorn@lumc.nl (R.v.D.); 2Scientific Bureau, Dutch Institute for Clinical Auditing, Rijnsburgerweg 10, 2333 AA Leiden, The Netherlands; m.wouters@nki.nl; 3Department of Ear-Nose-Throat ENT, Leiden University Medical Center, Leiden University, Albinusdreef 2, P.O. Box 9600, 2300 RC Leiden, The Netherlands; m.m.de_meza@lumc.nl; 4Department of Surgical Oncology, Netherlands Cancer Institute, Plesmanlaan 121, 1066 CX Amsterdam, The Netherlands; j.a.van_der_hage@lumc.nl; 5Department of Medical Oncology, Amsterdam UMC, VU University Medical Center, Cancer Center Amsterdam, 1081 HZ Amsterdam, The Netherlands; 6Department of Medical Oncology & Immunology, Netherlands Cancer Institute, Plesmanlaan 121, 1066 CX Amsterdam, The Netherlandsc.blank@nki.nl (C.U.B.); 7Department of Molecular Oncology & Immunology, Netherlands Cancer Institute, Plesmanlaan 121, 1066 CX Amsterdam, The Netherlands; 8Department of Medical Oncology, Leiden University Medical Centre, Albinusdreef 2, 2333 ZA Leiden, The Netherlands; 9Department of Medical Oncology, GROW School for Oncology and Developmental Biology, Maastricht University Medical Centre, P. Debyelaan 25, 6229 HX Maastricht, The Netherlands; mjb.essers.aarts@mumc.nl; 10Department of Medical Oncology, Zuyderland Medical Centre Sittard, Dr. H. van der Hoffplein 1, 6162 BG Sittard-Geleen, The Netherlands; 11Department of Medical Oncology, Radboud University Medical Centre, Geert Grooteplein Zuid 10, 6525 GA Nijmegen, The Netherlands; marye.boers-sonderen@radboudumc.nl; 12Isala Oncology Center, Isala, Dokter van Heesweg 2, 8025 AB Zwolle, The Netherlands; 13Department of Medical Oncology, University Medical Centre Groningen, University of Groningen, Hanzeplein 1, 9713 GZ Groningen, The Netherlands; g.a.p.hospers@umcg.nl; 14Department of Internal Medicine, Medisch Spectrum Twente, Koningsplein 1, 7512 KZ Enschede, The Netherlands; d.piersma@mst.nl; 15Department of Internal Medicine, Medical Centre Leeuwarden, Henri Dunantweg 2, 8934 AD Leeuwarden, The Netherlands; 16Department of Internal Medicine, Amphia Hospital, Molengracht 21, 4818 CK Breda, The Netherlands; mstevense@amphia.nl; 17Department of Medical Oncology and Radiology & Nuclear Medicine, Erasmus Medical Centre, ‘s-Gravendijkwal 230, 3015 CE Rotterdam, The Netherlands; a_van_der_veldt@erasmusmc.nl; 18Department of Internal Medicine, Maxima Medical Centre, De Run 4600, 5504 DB Eindhoven, The Netherlands; g.vreugdenhil@mmc.nl; 19Department of Biomedical Data Sciences, Leiden University Medical Centre, Einthovenweg 20, 2333 ZC Leiden, The Netherlands; 20Department of Medical Oncology, University Medical Center Utrecht, Utrecht University, Heidelberglaan 100, 3584 CX Utrecht, The Netherlands; k.suijkerbuijk@umcutrecht.nl

**Keywords:** melanoma, immune checkpoint inhibitors, side-effects

## Abstract

**Simple Summary:**

This nationwide cohort study evaluated the incidence and severity of immune-related adverse events (irAEs) in melanoma patients receiving adjuvant versus advanced anti-PD-1 therapy. A total of 1465 advanced melanoma patients and 908 adjuvant-treated patients were included. Patients in the adjuvant group were younger, had superior ECOG performance status, and had fewer comorbidities compared to those in the advanced group. No significant difference was observed in the incidence of grade III-IV irAEs between the two groups. However, higher ECOG status (>1) and the presence of any comorbidities were independently associated with an increased risk of irAE development across both treatment settings. Although adjuvant therapy was not linked to a heightened risk of severe irAEs compared to advanced therapy, it was more frequently discontinued due to toxicity in the adjuvant group. These findings provide critical insights for informing patient consultations regarding adjuvant anti-PD-1 treatment.

**Abstract:**

*Introduction*: The difference in incidence and severity of anti-PD-1 therapy-related adverse events (irAEs) between adjuvant and advanced treated melanoma patients remains unclear, as no head-to-head studies have compared these groups. *Methods*: This multi-center cohort study analyzed melanoma patients treated with anti-PD-1 in adjuvant or advanced settings between 2015 and 2021. Comorbidities and ECOG performance status were assessed before treatment, and grade III-IV irAEs were monitored during treatment. Univariate and multivariate regression analyses were conducted to identify factors associated with irAE development. *Results*: A total of 1465 advanced melanoma patients and 908 resected melanoma patients received anti-PD-1 therapy. Adjuvant-treated patients were younger, with a median age of 63 years compared to 69 years in the advanced group (*p* < 0.01), and had a better ECOG performance status (*p* < 0.01). Comorbidities were seen more frequently in advanced melanoma patients than in those receiving adjuvant treatment, 76% versus 68% (*p* < 0.01). Grade III-IV irAEs occurred in 214 (15%) advanced treated patients and in 119 (13%) adjuvant-treated patients. Multivariate analysis showed an increased risk of severe irAE development with the presence of any comorbidity (adjusted OR 1.22, 95% CI 1.02–1.44) and ECOG status greater than 1 (adjusted OR 2.00, 95% CI 1.20–3.32). Adjuvant therapy was not associated with an increased risk of irAE development compared to advanced treatment (adjusted OR 0.95, 95% CI 0.74–1.21) after correcting for comorbidities and ECOG performance score. Anti-PD-1 therapy was halted due to toxicity (any grade irAE) more often in the adjuvant setting than in the advanced setting, 20% versus 15% (*p* < 0.01). *Conclusions*: Higher ECOG performance status and presence of any comorbidity were independently associated with an increased risk of Grade III-IV irAE in adjuvant and advanced treated melanoma patients. Patients treated in the adjuvant setting did not have an increased risk of developing severe irAEs compared to advanced melanoma patients. These findings are of clinical significance in consulting patients for adjuvant anti-PD-1 treatment.

## 1. Introduction

Immune checkpoint inhibition therapy, particularly targeting the programmed death-1 (PD-1) pathway, has revolutionized the treatment landscape for melanoma, yielding significant improvements in patient outcomes across various disease stages. In patients with advanced melanoma (unresectable stage III-IV), the introduction of anti-PD-1 therapies has led to marked improvements in overall survival rates. Furthermore, in the adjuvant setting for melanoma patients who have undergone surgical resection of stage III-IV tumors, anti-PD-1 therapy has demonstrated a substantial benefit in enhancing local recurrence-free survival. By stimulating the immune system to target any residual microscopic disease, these therapies reduce the likelihood of melanoma recurrence after surgery [1,2,3,4,5,6,7]. Despite improved outcomes in the adjuvant and advanced setting, immune checkpoint inhibition can induce serious and long-lasting immune-related adverse events (irAEs) [8,9,10,11]. Immunotherapy-related adverse events can significantly impact the quality of life. Specific irAEs can occur, including fatigue, skin rashes, colitis, nephritis, hepatitis, nausea, and endocrine dysfunctions leading to discomfort, pain, and a reduced ability to perform daily activities, affecting both physical and mental well-being. Severe irAEs often necessitate the interruption or complete discontinuation of anti-PD-1 therapy. Managing irAEs typically require additional medications, such as corticosteroids (prednisone) or other immunosuppressive drugs. These medications can have their own side effects and complications, further burdening the patient and complicating their treatment regimen [9,12]. While the majority of adverse events tend to resolve over time with the administration of immunosuppressive medication, some side effects can be long-lasting [13]. This is particularly true for endocrine-related side effects, which often endure and can require ongoing management [9,14,15].

Interestingly, clinical trials investigating the efficacy of anti-PD-1 therapy in adjuvant and advanced melanoma patients have demonstrated a slightly higher incidence of treatment-related adverse events in patients receiving adjuvant therapy compared to those with advanced melanoma. This observation suggests that while anti-PD-1 therapy is generally effective across different stages of melanoma, the adverse event development risk profile may vary depending on the treatment setting [5,6,7,16]. Although this is true, it remains a matter of debate whether this holds true in the real-world setting. To date, only a conducted study by Verheijden and colleagues published in 2020 demonstrated a lower risk of severe irAEs in more advanced melanoma patients (stage IV M1c or higher) treated with immune checkpoint inhibitors compared to less extensive metastatic melanoma patients (stage IV M1a-b) [17]. The lower prevalence of immune therapy-related adverse events in advanced treated patients could be attributed to the fact that metastatic melanoma produces immune suppressive factors, damping the immune system [18,19]. 

This mechanism hypothesizes that patients with completely resected disease, and thus a more active immune system, could potentially develop adverse events more frequently in the adjuvant setting than patients with metastatic melanoma. The premise is that the robust immune response in these patients, which is essential for the success of adjuvant therapy, might also predispose them to a higher likelihood of immune-related adverse events. In contrast, patients with metastatic melanoma may have a more compromised immune system due to the advanced nature of their disease, which might reduce the incidence of these adverse events. Understanding this differential response is critical for tailoring immunotherapy regimens and managing potential side effects effectively in diverse patient populations. 

Despite these assumptions, no study has assessed and compared adverse events in real-world adjuvant and advanced anti-PD-1-treated patients. Additionally, potential confounders such as age, gender, and Eastern Cooperative Oncology Group (ECOG) performance status have not been analyzed and/or corrected for patients receiving treatment in the adjuvant versus advanced setting in daily clinical practice. Assessing these variables is necessary to identify potential factors associated with increased adverse event development, which can help optimize therapy decision-making [20]. For instance, patients with a history of autoimmune disease are more prone to develop disease-specific flare-ups and should be consulted for alternative treatment options (such as BRAF/MEK inhibition) or no anti-PD-1 treatment in the adjuvant setting [21,22]. 

In this multi-center nationwide register study using prospectively collected data, we assessed and compared demographic variables, ECOG status, comorbidities and type, duration, and severity of irAEs in melanoma patients receiving first-line adjuvant or advanced anti-PD-1 therapy.

## 2. Methods

### 2.1. Study Design

For this study, we utilized data from the Dutch Melanoma Treatment Registry (DMTR). The DMTR prospectively registers treated patients with advanced melanoma since 2012 and those receiving adjuvant therapy since 2018. Independent data managers, who undergo annual training, register the data, which is subsequently verified and reviewed by treating physicians to ensure high data quality [23].

This study was designed as a retrospective, observational cohort analysis and included all patients with advanced melanoma (irresectable stage III or IV) or resected stage III-IV melanoma who received anti-PD-1 monotherapy as their first-line treatment. Data were collected in the advanced patient group between January 2015 and September 2020 and from January 2018 to September 2020 in the adjuvant-treated patient group. The data set cutoff date was January 2021. The cutoff date of January 2021 provides a sufficient follow-up period to assess the outcomes and adverse events related to anti-PD-1 therapy. 

Patients were stratified according to treatment type (adjuvant versus advanced). Comorbidities were assessed prior to treatment initiation, providing a comprehensive baseline for each patient. Severe (grade 3 or higher) immune-related adverse events (irAEs) were meticulously recorded both during treatment and in cases where causality was suspected post-treatment, ensuring thorough monitoring of patient safety and treatment impact.

Moreover, the reason for therapy discontinuation was noted for every individual patient and included progressive disease, therapy completion, and therapy discontinuation due to toxicity (any grade irAE).

In compliance with Dutch regulations, the utilization of data from the Dutch Melanoma Treatment Registry (DMTR) for this research received approval from the Medical Ethics Review Committee of Leiden University Medical Center. This research was classified as exempt from the Medical Research Involving Human Subjects Act, meaning that no patient informed consent was required. 

### 2.2. Patients Characteristics

Registered demographic variables included age at diagnosis, gender, and Eastern Cooperative Oncology Group Performance Status (ECOG). Melanoma was staged according to the eighth edition of the American Joint Committee on Cancer (AJCC) melanoma staging system [24]. 

Comorbidities were documented prior to therapy initiation and consisted of neurological, cardiovascular, pulmonary, gastroenterological, urological, musculoskeletal, infectious, malignancy, rheumatological (cardiomyopathy, scleroderma, sarcoidosis, vasculitis), endocrine (thyroiditis, adrenal insufficiency), and inflammatory bowel disease (Crohn’s disease, colitis ulcerosa) comorbidities. The variable for preexisting autoimmune diseases encompassed rheumatological, endocrine, and inflammatory bowel disease comorbidities.

Immune therapy-related adverse events included grades 3 and 4, according to Common Terminology Criteria for Adverse Events (CTCAE) Version 4.0 IrAEs were categorized into the following groups: myelotoxicity, neuropathy, colitis, diarrhea, renal, lung, endocrine, fatigue, cutaneous, and hepatitis. Endocrine adverse events included adrenal insufficiency, thyroid disease, and hypophysitis. 

### 2.3. Statistical Analysis

Demographic variables were analyzed using descriptive statistics. Categorical variables were compared using the Pearson χ^2^ test, while continuous variables were assessed using the Wilcoxon rank-sum test. Univariate regression analysis was conducted to identify significant factors associated with the occurrence of adverse events. Factors identified as significant, along with treatment type, were included in the multivariate regression analysis. Statistical significance was set at two-sided *p*-values < 0.05. Data management and statistical analyses were performed using IBM SPSS Statistics version 24 (Armonk, NY, USA).

## 3. Results

### 3.1. Baseline Characteristics

A total of 1465 advanced melanoma patients received first-line anti-PD-1 therapy with a median age of 69 years (IQR 59–77), and the majority of patients were male (60%) (Table 1). The ECOG status was 0 in 818 patients (56%) and ≥1 in 573 patients (39%). Advanced melanoma patients were mostly staged as IV (89%), and irresectable stage III was seen less frequently (11%). An increased LDH level (>250 U/L) was observed in 366 advanced melanoma patients (25%), and 1083 (74%) of the patients had normal LDH levels (<250 U/L). These patients underwent an average of 11 immunotherapy cycles.

In the resected melanoma patient group, a total of 908 patients received adjuvant anti-PD-1 therapy. Adjuvant-treated patients had a median age of 63 years (IQR 54–72), and the majority of patients had an ECOG performance status of 0 (644 patients (71%)). Adjuvant-treated patients were staged IIIB or IIIC (81%) most frequently, while resected stage IV disease was seen less commonly (Iva, 9%; IVb, 3%; IVc, 1%), and these patients received an average of 9.2 immunotherapy cycles. The LDH values in this patient group were normal (<250 U/L) in the majority of cases (94%), and an elevated LDH value was seen in patients most frequently staged as stage IIIC (52%). 

Comparing advanced and adjuvant melanoma patients, the latter were younger (*p* < 0.01) and had an ECOG-performance status of 0 (*p* < 0.01) more frequently. No difference was observed in gender distribution (*p* = 0.52). Advanced melanoma patients received more mean cycles of immunotherapy than adjuvant-treated patients, 10.9 cycles versus 9.2 cycles (*p* < 0.01). Further, advanced melanoma patients had an increased LDH level more frequently than adjuvant-treated patients (*p* < 0.01), and these patients staged IVc the most frequently. 

Total median follow-up months, calculated from start to therapy up to the last in-patient clinic visit, was 11 months in the advanced melanoma group and 12 months in the adjuvant-treated group (*p* = 0.59).

### 3.2. Comorbidities

Advanced melanoma patients had any comorbidity in 1,128 (77%) cases (Table 2), with cardiovascular (23%) and neurological (17%) comorbidities being the most common. The incidence of other comorbidities (diabetes, pulmonary, gastroenterological, urological, musculoskeletal, infectious) ranged from 2% to 13%. 

In the adjuvant-treated group, 627 (69%) patients had any comorbidity, with cardiovascular (14%) and neurological (14%) comorbidities being the most frequent. Other comorbidities (diabetes, pulmonary, gastroenterological, urological, musculoskeletal, infectious) ranged from 2% to 10%. 

Comparing comorbidities between the treatment groups, any type of comorbidity (*p* < 0.01) and cardiovascular comorbidities (*p* = 0.02) were more prevalent in the advanced melanoma group. Other comorbidities, such as neurological, diabetes, pulmonary, gastroenterological, musculoskeletal, and infectious conditions, were similarly distributed between the two groups. Furthermore, a malignant comorbidity other than melanoma was observed more frequently in the advanced melanoma group (301 patients, 21%, versus 128 patients, 14%) (*p* < 0.01). 

Regarding any autoimmune comorbidities, no significant difference was observed between the two treatment groups, with 181 (13%) preexisting autoimmune comorbidities in advanced melanoma patients compared to 98 (11%) preexisting autoimmune comorbidities in adjuvant-treated patients (*p* = 0.25). Rheumatologic comorbidities were more common in advanced melanoma patients than in adjuvant-treated patients, 6% versus 4% (*p* = 0.03). Conversely, adjuvant-treated patients had endocrine comorbidities more frequently than advanced melanoma patients, 8% versus 6% (*p* = 0.01).

### 3.3. Characterization of Adverse Events

Any type of grade III-IV adverse event was seen in 214 (15%) advanced treated patients and in 119 (13%) adjuvant-treated patients (Table 3) (Figure 1), and this did not differ significantly between the two groups (*p* = 0.31). The incidence of specific irAEs (including myelotoxicity, neuropathy, colitis, renal, pulmonary, endocrine, fatigue, cutaneous, hepatitis, and other) was equally distributed among the two treatment groups, and the incidence ranged from 1 to 6%. 

Anti-PD-1 therapy was discontinued due to toxicity (any grade adverse event) more frequently in adjuvant-treated patients than in advanced treated patients (138 (20%) cases versus 196 (15%) cases (*p* < 0.01)). Additionally, 21 (1%) advanced melanoma patients died due to severe adverse events, whereas no adjuvant-treated patients died from therapy-related causes.

### 3.4. Univariate and Multivariate Analysis

The primary outcome of the univariate analysis was the occurrence of any type of treatment-related severe adverse event (grade III-IV). Univariate analysis demonstrated no significant association between age and gender for grade III-IV irAE development, with an OR of 1.00 (95% CI 0.99–1.01) (0.59) and 0.93 (95% CI 0.74–1.18) (*p* = 0.98), respectively (Table 4). Increased ECOG performance status (>1) and the presence of any type of comorbidity were associated with an increased risk of toxicity development, OR 2.03 (95% CI 1.23–3.34) (*p* = 0.01) and 1.22 (95% CI 1.03–1.44) (*p* = 0.02), respectively. Univariate analysis for treatment type demonstrated no increased risk of toxicity development for adjuvant therapy (with advanced therapy taken as reference), OR 0.88 (95% CI 0.69–1.12) (*p* = 0.67). The multivariate analysis included the significant variables ECOG, any comorbidity, and treatment type. 

Increased ECOG performance status (>1) and presence of any comorbidity remained statistically significant with an adjusted odds ratio of 2.00 (95% CI 1.20–3.32) (*p* = 0.01) and 1.22 (95% CI 1.02–1.44) (*p* = 0.01), respectively. Multiple variable analysis demonstrated no association between grade III-IV irAE development and type of therapy setting (advanced versus adjuvant), with an adjusted odds ratio of 0.95 (95% CI 0.74–1.21) (*p* = 0.39).

## 4. Discussion

To our knowledge, this is the first and largest population-based study to compare adverse events in adjuvant and advanced melanoma patients treated with anti-PD-1 therapy.

In our cohort, 212 (15%) advanced melanoma patients and 130 (13%) adjuvant-treated patients developed grade III/IV irAEs during median follow-up periods of 11 and 12 months, respectively. Multivariate analysis adjusted for ECOG status and comorbidities showed no increased risk of adverse event development in the adjuvant setting.

Adjuvant-treated patients were younger and generally healthier, with 71% of the patients having an ECOG status of 0, compared to 56% in the advanced group (*p* < 0.01). Cardiovascular comorbidities were seen less frequently in adjuvant-treated patients (*p* < 0.01), while endocrine comorbidities were more common (*p* = 0.01). Rheumatologic comorbidities were more frequent in advanced melanoma patients (*p* = 0.03). The observed difference in rheumatologic and endocrine comorbidity distribution might be due to the age difference in the treatment groups, as the incidence of rheumatologic comorbidities increases in older patients, and specific endocrine comorbidities (thyroid dysfunction) can be more prevalent in younger patients [25]. The total prevalence of preexisting autoimmune diseases was higher in advanced melanoma patients, albeit not significantly different (*p* = 0.25). 

To explain the observed clinical differences between the treatment groups: the need for anti-PD-1 therapy in the adjuvant setting may be less urgent, while in the advanced setting, advanced anti-PD-1 can be essential and might be initiated regardless of deteriorated ECOG performance status, higher age, or the presence of additional comorbidities. In addition, it is important to note that adjuvant immunotherapy has only been shown to improve recurrence-free survival, and no prolonged overall survival has yet been observed. Patients with poorer health and preexisting risk factors for adverse events are potentially less likely to be selected for adjuvant therapy, as the potential cons (adverse event development) may outweigh the benefits.

In our cohort, grade 3–4 irAEs were observed in 13% of adjuvant anti-PD-1 patients and 15% of advanced melanoma patients, with no significant difference between the groups (*p* = 0.31). A study by de Meza and colleagues, using the same data registry, reported a higher incidence of 18% in adjuvant-treated patients [26]. The higher incidence may be due to their longer follow-up period (median follow-up of 18 months vs. median follow-up of 12 months in our study), allowing more time for irAEs to develop. Previous studies have indeed shown that irAEs can occur after therapy cessation and that the incidence of irAEs can increase with a prolonged follow-up [9].

Albeit similar incidence of irAEs in both treatment groups, adjuvant-treated anti-PD-1 patients discontinued therapy due to toxicity (any grade irAE) more frequently than advanced treated patients. An explanation for the observed difference can be that anti-PD-1 treatment is preventive rather than curative. In contrast, advanced melanoma patients might continue treatment despite adverse events due to the critical nature of their therapy. This could result in a comparable overall toxicity profile, as early discontinuation in the adjuvant setting reduces prolonged exposure to toxicity.

Similar to our findings, clinical trials reported comparable incidences of grade 3–4 irAEs with adjuvant nivolumab. The CheckMate 238 trial for adjuvant-treated stage III melanoma (median follow-up of 18 months) reported 14%, and the CheckMate 76K trial for adjuvant-treated stage II melanoma (median follow-up of 12 months) reported an incidence of 10%. For advanced melanoma, the CheckMate 066 trial (median follow-up of 17 months) showed a 12% incidence. The KEYNOTE-054 study on adjuvant pembrolizumab for resected stage III melanoma reported 15%, while the KEYNOTE-006 study on pembrolizumab for advanced melanoma reported a 13% incidence [2,9,11].

The onset of irAEs is multifactorial. Established clinical factors that contribute to irAE development in advanced melanoma patients treated with immune checkpoint inhibitors include a deteriorated ECGO status and preexisting autoimmune diseases [12,20,27,28]. In addition, the extent of metastatic disease might dampen the immune system activity and could thereby potentially reduce the irAE development in advanced melanoma patients [18,19,29]. To elucidate this, melanoma tumors can evade immune-mediated destruction through immunosuppressive mechanisms that inhibit T-cell activation. However, some tumors still exhibit high levels of CD8+ T cells despite being suppressed by tumoral factors. This elevated T-cell level may result from innate immunosuppressive mechanisms such as indoleamine-2,3-dioxygenase (IDO), PD-L1/B7-H1, tryptophan 2,3-dioxygenase (TDO), and FoxP3+ regulatory T cells [30,31,32]. These pathways are driven by the innate immune system rather than activated by tumor cells.

Interestingly, the expression of IDO and TDO is associated with reduced tumor-infiltrating immune cells and poor responses in malignancies. Furthermore, the expression of IDO and/or TDO could potentially decrease the efficacy of immune checkpoint inhibition, making these mechanisms significant targets for addressing melanoma patients who do not respond to anti-PD-1 therapy [33]. Moreover, a recently published study demonstrated that the molecule heme plays a role in the activation of IDO and TDO and, therefore, could potentially be a therapeutic target as well [34,35]. 

This tumoral immune response remains a complex, multifactorial challenge. Patients with widespread metastatic cancer may be exposed to higher levels of immunosuppressive factors than patients rendered disease-free. Despite this, the innate immune system plays an important role, which might still induce a similar frequency of adverse event development in metastatic melanoma patients as compared to adjuvant-treated patients.

In our multivariate analysis, we assessed the effect of therapy setting (adjuvant versus advanced) and corrected for factors associated with an increased risk of irAE development (ECOG performance status and comorbidities). We did not show an increased risk of adverse event development in the adjuvant setting. Potential selection bias may influence our results, as patients with a higher tumor load might be more frequently selected for anti-CTLA-4/anti-PD-1 combination therapy rather than anti-PD-1 monotherapy. Despite this, anti-PD-1 treatment can induce an immune response, irrespective of the disease stage. In addition, adjuvant-treated patients may have microscopic residual disease, resulting in a vigorous immune response similar to that seen in patients with detectable metastatic disease. The uniform mechanism of action may potentially lead to similar incidences of immune-related adverse events (irAEs) across different stages of melanoma.

Our findings indicate that ECOG performance status and the presence of any comorbidity are independently associated with an increased risk of adverse event development in both adjuvant and advanced melanoma patients treated with anti-PD-1 therapy. Patients with these preexisting risk factors for adverse event development should receive thorough counseling on whether to pursue adjuvant anti-PD-1 therapy initiation, especially since no improvement in overall survival has been reported for melanoma patients treated with adjuvant anti-PD-1.

The limitations of our study include that the data registry only captures grade 3–4 irAEs and the retrospective nature of the study. Despite these limitations, our data were collected prospectively by independent data managers and reviewed by treating physicians. Also, this is the largest real-world study to date investigating grade III-IV irAEs, with extensive descriptions of patient comorbidities.

## 5. Conclusions

In this real-world study, our data demonstrated that an ECOG score > 1 and the presence of any type of comorbidity were associated with an increased risk of immune therapy-related adverse event development in resected and irresectable stage III-IV melanoma patients undergoing immunotherapy. Anti-PD-1 treatment in the adjuvant setting, compared to the advanced setting, was not significantly associated with an increased risk of grade III-IV adverse event development. Anti-PD-1 therapy was halted due to toxicity (any grade irAE) more frequently in the adjuvant setting, potentially as treatment might be less essential as compared to advanced anti-PD-1-treated melanoma patients. These findings hold clinical importance in advising patients about the potential benefits of adjuvant anti-PD-1 therapy.

## Figures and Tables

**Figure 1 cancers-16-02656-f001:**
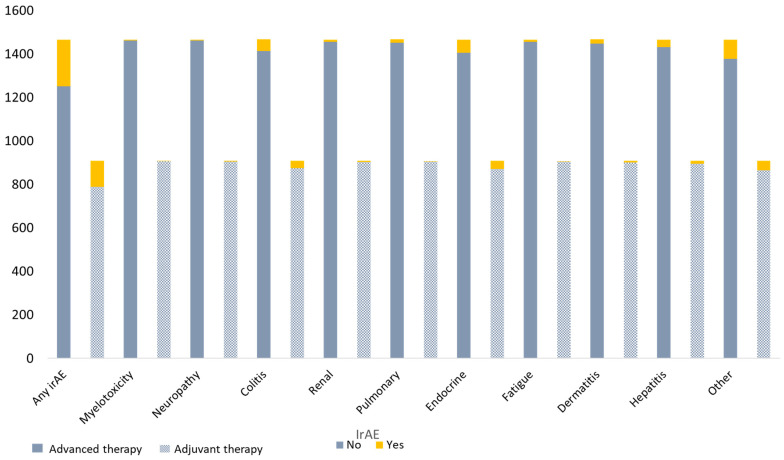
Types of grade III-IV adverse events in advanced and adjuvant melanoma patients treated with anti-PD-1 therapy.

**Table 1 cancers-16-02656-t001:** Patient characteristics.

	Advanced (N = 1465)	Adjuvant (N = 908)	*p*-Value
**Characteristics**			
**Age at diagnosis**			<0.01
Median (interquartile range), years	69 (59–77)	63 (54–72)	
**Sex**			0.52
Male	882 (60)	523 (58)	
Female	583 (40)	385 (42)	
**ECOG performance status**			<0.01
0	818 (56)	644 (71)	
1	482 (33)	203 (22)	
>1	91 (6)	14 (2)	
Unknown	74 (5)	47 (5)	
**AJCC, 8th edition, stage**			<0.01
IIIA	NA	54 (6)	
IIIB	NA	319 (35)	
IIIC	158 (11)	413 (46)	
IIID	NA	12 (1)	
IVa	236 (16)	79 (9)	
IVb	385 (26	30 (3)	
IVc	483 (33)	1 (1)	
IVd	202 (14)	NA	
Unknown	1 (1)	NA	
**LDH levels**			
Normal (<250 U/L)	1083 (74)	850 (94)	<0.01
Increased (>250 U/L)	366 (25)	33 (4)	
Missing	16 (1)	23 (3)	
**Total mean cycles**			
	10.9	9.20	<0.01

**Table 2 cancers-16-02656-t002:** Prevalence of comorbidities among study participants.

		Advanced (N = 1465)	Adjuvant (N = 908)	*p*-Value
**Any comorbidity**	no	337 (23)	281 (31)	**<0.01**
	yes	1128 (77)	627 (69)	
**Neurological**	no	1215 (83)	780 (86)	0.34
	yes	250 (17)	128 (14)	
**Cardiovascular**	no	1135 (77)	779 (86)	**<0.01**
	yes	330 (23)	129 (14)	
**Diabetes**	no	1286 (88)	811 (90)	0.61
	yes	179 (12)	94 (10)	
**Pulmonary**	no	1300 (89)	819 (90)	0.38
	yes	165 (11)	89 (10)	
**Gastroenterological**	no	1313 (90)	839 (92)	0.19
	yes	152 (10)	69 (8)	
**Urological**	no	1303 (89)	834 (92)	0.13
	yes	162 (11)	74 (8)	
**Musculoskeletal**	no	197 (13)	806 (89)	0.13
	yes	1268 (87)	102 (11)	
**Infectious**	no	1426 (98)	906 (98)	0.74
	yes	29 (2)	16 (2)	
**Malignancy**	no	1164 (79)	780 (86)	**<0.01**
	yes	301 (21)	128 (14)	
**Any autoimmune disease**	no	1284 (87)	810 (89)	0.25
	yes	181 (13)	98 (11)	
**IBD**	no	1441 (98)	900 (99)	0.12
	yes	24 (2)	8 (1)	
**Rheumatologic**	no	1371 (94)	869 (96)	**0.03**
	yes	94 (6)	39 (4)	
**Endocrine**	no	1381 (94)	832 (92)	**0.01**
	yes	84 (6)	76 (8)	

**Table 3 cancers-16-02656-t003:** Type of severe/grade 3+ adverse events and presence or absence in anti-PD-1-treated advanced and adjuvant melanoma patients.

		Advanced (N = 1465)	Adjuvant (N = 908)	*p*-Value
Any type of adverse event	No	1251 (85)	789 (87)	0.31
	Yes	214 (15)	119 (13)	
Myelotoxicity	No	1462 (1)	907 (99)	0.66
	Yes	3 (1)	1 (1)	
Neuropathy	No	1462 (99)	905 (99)	0.41
	Yes	3 (1)	3 (1)	
Colitis	No	1414 (96)	875 (96)	0.67
	Yes	53 (4)	33 (4)	
Renal	No	1455 (99)	903 (99)	0.62
	Yes	10 (1)	5 (1)	
Pulmonary	No	1451 (99)	904 (99)	0.25
	Yes	16 (1)	4 (1)	
Endocrine	No	1405 (96)	870 (96)	0.92
	Yes	60 (4)	38 (4)	
Fatigue	No	1456 (99)	904 (99)	0.14
	Yes	9 (1)	2 (1)	
Cutaneous	No	1448 (99)	901 (99)	0.27
	Yes	19 (1)	7 (1)	
Hepatitis	No	1432 (98)	895 (98)	0.10
	Yes	33 (2)	13 (1)	
Other	No	1377 (94)	864 (95)	
	Yes	88 (6)	44 (5)	

**Table 4 cancers-16-02656-t004:** Uni- and multivariate regression analysis assessing clinical factors associated with irAE occurrence.

	Univariate (OR, 95% CI)	Multivariate (OR, 95% CI)
Age	1.00 (0.99–1.01)	
Gender		
Male	Reference	
Female	0.93 (0.74–1.18)	
ECOG		
0–1	Reference	
>1	2.11 (1.23–3.49)	2.00 (1.20–3.32)
Comorbidity		
Absent	Reference	
Any	1.24 (1.04–1.46)	1.22 (1.02–1.44)
Treatment Type		
Advanced	Reference	
Adjuvant	0.88 (0.69–1.12)	0.95 (0.74–1.21)

## Data Availability

The data presented in this study are available upon request from the corresponding author. The data are not publicly available because of the protection of privacy.

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
