# Peer review of "Adverse Events in Anti-PD-1-Treated Adjuvant and First-Line Advanced Melanoma Patients"

_cancers, 2024, doi:10.3390/cancers16152656_

Round 1

Reviewer 1 Report

Comments and Suggestions for Authors

Overall an interesting manuscript with novel data. My only concern: the authors must discuss the important role of IDO1 and TDO in melanoma either in the introduction or discussion section. There are several previous reports like  PMID: 23986400, 10.1007/s13555-019-0292-3, 10.4161/2162402X.2014.982382 which discusses their roles in melanoma along with PD-1. Also recently 2 manuscripts were published which discovered the heme insertion mechanisms into IDO1 and TDO, 10.1016/j.freeradbiomed.2022.01.008, 10.1016/j.jbc.2023.104753 which must also be mentioned in the manuscript. All these mentioned references must be included in the paragraph where the authors would discuss the important roles of IDO1 and TDO in melanoma anti-PD-1 therapy. This would greatly enrich the manuscript by providing necessary background and also discuss the roles of these 2 important enzymes along with PD-1 in melanoma. Thank you.

Author Response

Thank you for these comments. We have adjusted our text accordingly and included the suggested references, to create a broader view on the immune response background of melanoma. 

Reviewer 2 Report

Comments and Suggestions for Authors

dear authors ,

I suggest minor revision.

Could you detail LDH level with the stage ( III or IV ....? ) in table 1 and also in the text? 

thank you

Author Response

Comments 2: 

dear authors ,
I suggest minor revision.
Could you detail LDH level with the stage ( III or IV ....? ) in table 1 and also in the text? 
Thank you. 

Response: 
Thank you for this valuable notion. We have adjusted our text accordingly. 

Reviewer 3 Report

Comments and Suggestions for Authors

The paper "Adverse events in anti-PD1 treated adjuvant and first-line advanced melanoma patients" presents a thoroughly designed and well-executed study that offers valuable insights into a “hot topic” in oncology field and melanoma treatment. 

It addresses an important gap in current knowledge by comparing the incidence and severity of anti-PD-1 therapy-related adverse events (irAEs) in adjuvant and advanced melanoma patients. This is a critical area of research, as understanding these differences can guide clinical decision-making and patient management.

It is also crucial to meticulously select the patients who can potentially benefit from this type of therapy, considering the associated adverse effects.

The multi-center cohort design enhances the study's robustness and generalizability. Also by including a large sample by access to national records, the researchers ensured sufficient statistical power to detect meaningful differences and associations. This extensive dataset allows a more reliable and comprehensive analysis. The paper is clear and well-written allowing the reader to clearly understand the research process, findings, their clinical significance and implications.

In conclusion, this study is a significant contribution to melanoma treatment and i would suggest to be accepted in present form. 

Author Response

Thank you for this comment. Hopefully our study results will influence daily practice in the selection for adjuvant therapy consideration in melanoma patients.